# Regulation of Platelet Production and Life Span: Role of Bcl-xL and Potential Implications for Human Platelet Diseases

**DOI:** 10.3390/ijms21207591

**Published:** 2020-10-14

**Authors:** Emma C. Josefsson, William Vainchenker, Chloe James

**Affiliations:** 1The Walter and Eliza Hall Institute of Medical Research, Melbourne, VIC 3052, Australia; josefsson@wehi.edu.au; 2Department of Medical Biology, University of Melbourne, Melbourne, VIC 3052, Australia; 3University Paris-Saclay, INSERM UMR 1270, Gustave Roussy, 94800 Villejuif, France; william.vai@wanadoo.fr; 4University of Bordeaux, INSERM U1034, Biology of Cardiovascular Diseases, 33600 Pessac, France; 5Laboratory of Hematology, Bordeaux University Hospital Center, Haut-Leveque Hospital, 33604 Pessac, France

**Keywords:** platelet, megakaryocyte, Bcl-xL

## Abstract

Blood platelets have important roles in haemostasis, where they quickly stop bleeding in response to vascular damage. They have also recognised functions in thrombosis, immunity, antimicrobal defense, cancer growth and metastasis, tumour angiogenesis, lymphangiogenesis, inflammatory diseases, wound healing, liver regeneration and neurodegeneration. Their brief life span in circulation is strictly controlled by intrinsic apoptosis, where the prosurvival Bcl-2 family protein, Bcl-xL, has a major role. Blood platelets are produced by large polyploid precursor cells, megakaryocytes, residing mainly in the bone marrow. Together with Mcl-1, Bcl-xL regulates megakaryocyte survival. This review describes megakaryocyte maturation and survival, platelet production, platelet life span and diseases of abnormal platelet number with a focus on the role of Bcl-xL during these processes.

## 1. Platelets

Platelets are small (2–4 µm in diameter), anucleate blood cells with a characteristic discoid shape. They have multiple functions and a life span of 7–10 days in human. They are produced by megakaryocytes (MKs), mostly in the bone marrow. Around 10^11^ platelets are produced every day. The main function of platelets is to participate in the physiological process of haemostasis, allowing bleeding to stop at the site of an injury, while maintaining normal blood flow elsewhere in the circulation. The inner surface of blood vessels is covered by endothelial cells that act as anticoagulant. After a vessel injury, the subendothelial matrix is exposed to the blood and several components of the blood will be allowed to interact with this matrix. Among these blood components are platelets. Platelets are present at 150 to 400 millions per millilitre (mL) of blood. They express receptors on their cell surface, but in a healthy blood vessel, and under normal blood flow, these receptors do not meet their ligands, and platelets do not adhere to surfaces or to each other. However, when the subendothelial matrix is exposed, the ligands for some of the platelet receptors are revealed and platelets adhere. The most important pathways in platelet adhesion is through the binding of immobilised von Willebrand factor (VWF) in the subendothelium to platelet GPIb-IX-V (CD42) and subendothelial collagen to platelet GPVI. The binding of platelet receptors with their ligands leads to platelet activation, i.e., secretion of the content of their granules, production of thromboxane A_2_ and subsequent activation of other local platelets, exposure of membrane phospholipids and induction of conformational changes of the αIIbβ3 (CD41/CD61) receptors. These receptors, when activated, can bind the fibrinogen that physiologically circulates in the blood. Activated platelets will thus aggregate with each other thanks to bridges made of fibrinogen and activated αIIbβ3 receptors on the platelet surface. Within seconds, a platelet plug will be formed allowing the bleeding to stop. This process is called primary haemostasis. In parallel, another process occurs after exposure of tissue factor in the subendothelial matrix that is called coagulation. Coagulation allows formation of fibrin that will stabilise the platelet clot (for review see [1]).

Platelets also have other roles than participating in haemostasis. They have been shown to be involved in immunity, antimicrobial defense [2], cancer growth and metastasis [3,4,5], tumour angiogenesis [6], lymphangiogenesis [5,7], inflammatory diseases [8,9,10], wound healing [11], liver regeneration [12] and neurodegeneration [13].

The spleen and the liver are the organs where aged platelets are destroyed, if they have not been used for haemostasis after 10 days of life. Moreover, not all platelets are circulating in the blood flow, but approximately 30% of the total platelet mass exists as an exchangeable pool residing in the spleen. This explains why increased spleen mass (splenomegaly) can be associated with lower platelet count and why absence of spleen (asplenia) is associated with increased platelet counts.

## 2. Regulation of Platelet Life Span by Apoptosis: Role of Bcl-xL

Intrinsic apoptosis is tightly controlled by the Bcl-2 family of proteins, functionally divided into pro- and antiapoptotic proteins. Bak and Bax are the proapoptotic molecules that dictate cell fate. Activation of Bak and Bax will cause irreversible damage to the mitochondrial outer membrane with subsequent cytochrome *c* release, apoptosome assembly and caspase activation. In healthy cells, the antiapoptotic members Bcl-xL, Bcl-2, Mcl-1, Bcl-w, A1 and Bcl-b [14] restrain Bak and Bax. In response to cellular stress, a third group of Bcl-2 proteins, the BH3-only initiator proteins (Bim, Puma, Bad, Noxa, Bik, Hrk and Bmf), can inhibit the prosurvival proteins and indirectly as well as directly contribute to the activation of Bak and Bax [15,16,17,18]. The balance between pro- and antiapoptotic members of the intrinsic apoptosis pathway varies depending on cell type.

In adults, Bcl-xL plays an essential survival function in platelets [19,20]. Platelets are a unique model in terms of life span regulation because they mostly depend on only two antagonistic Bcl-2 family members, i.e., Bcl-xL for their survival and Bak for their death [19,21], when in circulation at steady-state. However, in response to apoptotic triggers, such as pharmacological inhibition or genetic deletion of Bcl-x_L_, not only Bak, but both Bak and Bax regulate platelet lifespan [19,22,23,24].

How Bcl-xL/Bak controls platelet life span has been under intense scrutiny: in 2007, the Kile laboratory found mutations in *Bcl-x* that resulted in decreased platelet counts in mice due to shortened platelet life span [19]. Conversely, *Bak*-deficient mice exhibited a marked augmentation in platelet count due to increased platelet half-life [19,22]. Pharmacological inhibition of Bcl-xL causes thrombocytopenia in mice and humans, first demonstrated with drugs such as navitoclax (ABT-263) and ABT 737 [19,20,25]; however, in addition to Bcl-xL, they also inhibited prosurvival Bcl-2 and Bcl-w.

Do other prosurvival Bcl-2 family members, in addition to Bcl-xL, contribute to the regulation of platelet life span? There is compelling evidence that both human and murine platelets express Bcl-2 protein [19,20,22,26,27,28,29,30,31,32]. However, as pharmacological Bcl-2 inhibition [33,34] or specific deletion in MKs and platelets [27] had no effect on platelet number and life span, Bcl-2 is unlikely to play a role in the regulation of platelet life span. While Bcl-w has been shown to be present in platelets [20,28], it is not likely to control platelet life span since systemic *BCL2L2* knockout mice did not alter platelet numbers [35,36]. It is currently unclear if A1 protein is present in platelets, but A1 RNA has been detected in young human and murine platelets [37,38]. In mice, platelet counts were not changed by systemic *A1a* knock out [39] and another study targeting all A1 isoforms did not report any alteration of platelet counts [40]. The Mcl-1 protein has a short half-life in part as a result of fast proteosomal degradation. While Mcl-1 is critical for haematopoietic stem cells and multipotent haematopoietic progenitors [41] (Figure 1), neither haematopoietic overexpression of Mcl-1 nor conditional deletion in the platelet lineage affected platelet counts or platelet life span in mice [42,43,44]. Mcl-1 protein is present in MKs, but absent in both human and murine platelets [19,20,22,28,42,43]. However, young platelets could potentially express some Mcl-1 as it was detected in murine platelets following proteasome inhibitor treatment in vivo [43]. As specified above, anuclear platelets can undergo intrinsic apoptosis; however, their incapability to uphold Mcl-1 protein levels makes them different to most other cell types, as they are dependent on prosurvival Bcl-xL only [45,46,47,48]. Our current knowledge is therefore that platelet life span in vivo is dictated by the balance between prosurvival Bcl-xL and proapoptotic Bak (Figure 1).

To explain how platelets die, the hypothesis of a differential life span between Bcl-xL and Bak was proposed under the name of the “molecular clock model” [19]: as platelets have limited capacity to synthesise new proteins, if Bcl-xL has a shorter life span than Bak, then Bak molecules eventually outnumber Bcl-xL and platelet apoptosis ensues (Figure 2). In further support, in vitro aged human platelets at 37 °C exhibited Bcl-xL degradation [26]. However, in 2013 the Kile laboratory revised their molecular clock model and showed that in vivo aged platelets and young ones contain the same amount of Bcl-xL [49]. If, therefore, platelet in vivo apoptosis is not simply the result of Bcl-xL degradation over time, two hypotheses remain on what triggers platelet apoptosis in vivo (Figure 2): (1) a direct activation of Bak by an unknown factor or (2) the function of Bcl-xL is modified in aged platelets that may induce a decreased inhibitory activity on Bak. Furthermore, studies have shown that cell surface glycoproteins are modified in aged platelets leading to their recognition and clearance by liver macrophages and hepatocytes [50,51]. Future studies are needed to determine if there is a link between platelet age-dependent changes in glycosylation and platelet apoptosis or if these processes are distinct.

In regard to the first hypothesis, Bak activation could occur by BH3-only pro-death proteins. So far, only genetic deletion of Bad has shown an effect, although minor, on platelet life span [52], while deletion of Bid and Bim [28] or Puma [38] had no impact. It therefore remains to be established whether other BH3-only proteins play a critical role in induction of apoptosis in platelets. Moreover, the second hypothesis remains to be tested, i.e., whether the activity of Bcl-xL is modified as platelets age, and whether this modification would lead to impairment of proapoptotic Bak restriction by Bcl-xL. It may be that the ubiquitine–proteasome system participates in this process, as platelets express standard and immunoproteasome.

## 3. Megakaryopoiesis and Its Regulation

Platelets are produced by their precursors, MKs, in a process called megakaryopoiesis. This differentiation process occurs mainly in the bone marrow, but recent murine data have described that it may also occur in the extravascular space of the lungs [53]. The MK/platelet lineage is derived from a haematopoietic stem cell (HSC), which gives rise to all haematopoietic lineages including the lymphoid and myeloid lineages. This HSC is located at the top of the hierarchy and in the classical model this HSC will progressively lose its self-renewal capacities, a property absolutely required for an HSC and then after a multi stage process will commit to one lineage as the MK/Platelet lineage by giving a restricted MK progenitor [54]. In this model, there is strong evidence that the MK lineage derives from a bipotent erythroid/MK progenitor. Recent results also suggest that MK progenitors can directly derive from the HSC and that some HSC are biased towards the MK/platelet lineage [55]. It has been suggested that up to 60% of the MKs arise directly from a MK-biased HSC [56]. The MK progenitors are able to proliferate and will switch from mitosis to endomitosis, an incomplete mitosis related to a failure in cytokinesis leading to a polyploid cell associated with an increase in cell size. This polyploidisation leads to cells with a 2 N ploidy ranging from 2 N to 128 N, with a modal ploidy at of 16 N [57]. Because platelets are formed from the MK cytoplasm, this process leads to a marked amplification in platelet production. As soon as MK enters the polyploidisation process, it begins to synthesise the main platelet proteins, such as αIIb, GPIb and VWF, and develop specific organelles such as the demarcation membrane system (DMS), a highly tortuous invaginated membrane system necessary for the formation of platelets.

The later stage of megakaryopoiesis is often referred to as proplatelet formation. It corresponds to the stage when usually large polyploid MKs shed long-branching cytoplasmic protrusions called proplatelets into the sinusoidal blood vessels from the bone marrow [58]. These proplatelets arise from the DMS, unwinding the DMS under the driver forces of the cytoskeleton. With the shear stress in blood flow, proplatelet fragments are teared off and will fragment into platelets in the blood stream [59]. However, a recent 2020 study by Potts et al., proposes that MK membrane budding, rather than proplatelet formation, supplies the majority of the platelet biomass without induction of cell death [60]. Even so, platelets are small cytoplasmic fragments of MK cytoplasm, devoid of nucleus and able of residual protein synthesis due to the presence of mRNA coming from the MK. Alternatively, entire MK may migrate in the blood flow where they could be fragmented predominantly in the small lung vessels.

Thrombopoietin (TPO) is the main extrinsic regulator of platelet production [61] and binds to its cognate receptor, called MPL, which is a type 1 homodimeric cytokine receptor. Binding of TPO to MPL induces signalling through JAK2 and then downstream effectors such as STAT5, which regulates Bcl-xL expression [62,63]. In humans, homozygous loss of function mutations of MPL or TPO leads to a profound thrombocytopenia followed by an aplastic anemia, underscoring the role of the TPO/MPL axis in platelet production. TPO behaves as a hormone and is mainly synthesised by the liver. The regulation of the TPO plasma level is mainly, but not uniquely, regulated by its clearance. Indeed, platelets express MPL, bind TPO, and then TPO is endocytosed and degraded by platelets inversely correlating the blood TPO and platelet levels [64].

Intrinsically, MK differentiation is regulated by several transcription factors that can be in the same transcriptional large complex. The most important transcription factors are GATA1, RUNX1, FLI-1, GFI-1b and TAL1/SCL, some of them such as GATA1 regulating Bcl-xL expression. The transcription factor NF-E2 is involved in late stages of differentiation [65].

## 4. Diseases of Platelet Numbers in Humans

The normal platelet count ranges from 150 to 400 × 10^6^ per mL in humans. The platelet level is affected in numerous diseases and can increase over 400 × 10^6^/mL (thrombocytosis) or decrease to less than 150 × 10^6^/mL (thrombocytopenia). Platelet disorders can be totally indolent. Indeed, thrombocytopenia leads to spontaneous bleeding only when the platelet number is less than 30 × 10^6^/mL. On the contrary, thrombocytosis, in certain settings, can be associated with thrombosis, but the risk of thrombosis is not correlated with the number of platelets. Therefore, routine screening identifies many patients with asymptomatic thrombocytosis or thrombocytopenia.

### 4.1. Thrombocytosis (Platelet Count over 400 × 10^6^/mL)

Most thrombocytosis (over than 90%) are reactive and secondary to infection or inflammatory diseases (autoimmune, cancer and tissue injury) as inflammatory cytokines increase platelet production directly or indirectly by increasing TPO synthesis (Table 1). Iron deficiency can also lead to moderate thrombocytosis, or even thrombocytopenia when the iron deficiency is severe. The precise mechanism is unknown although it has been suggested that iron deficiency induces a bias towards MK commitment of a bipotent erythroid/MK progenitor [66]. Another setting associated with reactive thrombocytosis is when the spleen has been removed (splenectomy) or is not functional (asplenism).

Non-reactive thrombocytosis can be either inherited or clonal. In this latter case, they are due to the acquisition of somatic mutations in a haematopoietic stem cell, giving rise to myeloproliferative neoplasms (MPN) or myeloproliferative/myelodysplastic disease. Interestingly, whether they are inherited or acquired they are related to abnormalities in the TPO/MPL/JAK2 axis [62]. Inherited thrombocytosis are very rare. They can be due to mutations in the *TPO* gene (with increased TPO production), *MPL* (with a defect in TPO clearance), and rarely of *JAK2*. However, there also some unexplained inherited thrombocytosis.

Acquired, clonal thrombocytosis is mainly found in BCR-ABL-negative MPN. This disease includes three disorders, polycythemia vera, essential thrombocythemia and primary myelofibrosis, which are characterised by increased blood cell production. Most of them are driven by mutations in three genes: *JAK2* (JAK2V617F), *CALR* (calreticulin) and *MPL* [67]. All these mutations induce a constitutive activation of the MPL/JAK2 axis, and thus increased expression of Bcl-xL [68]. Nevertheless, the molecular origin of 10% to 20% of essential thrombocythemia and primary myelofibrosis is still unknown; these are called triple-negative MPN (JAK2V617F-CalR- and MPL-negative). Given the common clinical and biological features between JAK2V617F and triple-negative MPN, it is very probable that the same common signalling pathways are activated, i.e., the MPL/JAK2 axis [67].

### 4.2. Thrombocytopenia (Platelet Count Less Than 150 × 10^6^/mL)

Thrombocytopenias are frequent, but in most cases they are not isolated but rather associate with acute or complex pathologies. They are related either to a defect in platelet production, increased platelet destruction or consumption, abnormal pooling or eventually to mixed mechanisms (Table 2). Thrombocytopenia can be acute or chronic, acquired or inherited as reviewed recently [69].

#### 4.2.1. Thrombocytopenia with Abnormal Pooling

They are the consequence of increased spleen sequestration (hypersplenism) that can occur when the spleen is enlarged (splenomegaly).

#### 4.2.2. Thrombocytopenia due to Defective Platelet Production

Thrombocytopenias that are related to a defect in platelet production are mostly the consequences of a bone marrow infiltration, most often due to the proliferation of malignant cells, such as leukemic cells or cancer cells. They are thus most often associated with diminution of other blood cell types (i.e., anaemia and leukopenia). They can also be the consequence of a defect in MK differentiation. That is observed in myelodysplastic syndrome, vitamin B12 or folate deficiency. Here again, the thrombocytopenia is rarely isolated.

Thrombocytopenias related to a defect in platelet production are frequently observed in inherited thrombocytopenia [70]. These latter are due to various defects: (i) nearly absence of MKs in the rare cases of MPL or TPO mutations; (ii) defects in MK differentiation due to mutations in transcription factors (FLI1, RUNX1, GATA1, ETV6, IKZF5); and (iii) defects in proplatelet formation due to mutations in genes coding for proteins involved in the cytoskeleton such as GPIB, Filamin A, MYH9, Actinin B, and DIAPH1. The latter are frequently characterised by macrothrombocytopenia. However, even if inherited thrombocytopenias are increasingly understood, the molecular basis is still unknown for a significant number of patients.

#### 4.2.3. Thrombocytopenia due to Increased Platelet Destruction or Consumption

Thrombocytopenias due to increased platelet destruction are heterogeneous disorders. They include immunological and non-immunological processes. Non-immunological processes of platelet destruction are mostly due to platelet consumption as in (i) microangiopathies, which are associated with anemia and schizocytes; (ii) large haemorrhages; (iii) disseminated intravascular coagulation; and (iv) sepsis due to viral or bacterial infections. Immunological mechanisms of platelet destruction are mostly due to an autoimmune process called immune thrombocytopenia purpura (ITP) [71]. It is usually related to auto-antibodies directed against the main platelets glycoproteins αIIbβ3 or GPIb-IX-V. This immune process is usually idiopathic, but can be also secondary to some autoimmune disorders such as systemic lupus erythematosus, lymphoproliferative diseases or immune deficiency. The thrombocytopenia is related to enhanced platelet destruction in the spleen and/or the liver and therefore a very short platelet life span. However, a defect in platelet production is also frequently observed. It is presumably related to an immune destruction of MKs in the bone marrow. The main diagnostic problem of ITP is that the current techniques to detect anti-platelet antibodies are not sensitive or specific enough. Thus, ITP remains a diagnosis of exclusion.

Thus, it is likely that a part of what is diagnosed as ITP, and more particularly ITP that are resistant to steroid therapy or immune suppression, is actually not related to an immune process. It is clear that some inherited thrombocytopenias are misdiagnosed as ITP. However, the diagnosis will be facilitated by the development of Next-Generation Sequencing targeted on genes involved in inherited thrombocytopenia. Furthermore, there are patients with acquired chronic thrombocytopenias who are diagnosed with ITP, but do not have any immunological features but rather a real defect in MK differentiation. It is possible that these patients do not have an immune disease, but rather a haematological malignancy. It is conceivable that at least some of these patients can have an acquired defect in MK differentiation and/or a short platelet life span. In these cases, the phenotype (bone marrow aspiration, measurement of platelet life span) could be close to what is observed in typical ITP.

## 5. Role of Bcl-xL in Megakaryocyte Survival

Most of the data available on the role of Bcl-xL in MK life span have been obtained in mice. Similar to platelets, MK survival is regulated by Bcl-xL, but a major difference is that they also depend on prosurvival Mcl-1 (Figure 1). Cultured murine mature MKs were shown to undergo apoptosis and failed to produce proplatelets after genetic loss of *BCL2L1 gene (Bcl2-like1 or Bcl-x)*, or in response to pharmacological Bcl-xL inhibition [22]. In mice, deletion of *BCL2L1* in the MK lineage led to severe macrothrombocytopenia, associated with the release of abnormal vacuolar fragments in the bone marrow sinusoids [22]. It is therefore likely that MKs at the latest stage of their maturation process depend on Bcl-xL in order to properly progress through the process of platelet release. Nevertheless, in vivo loss of Bcl-xL did not negatively alter growth and maturation of MKs [22]. This finding suggested redundancy with other pro-survival proteins, and strikingly, genetic dual loss of *Mcl-1* and *BCL2L1* in the MK lineage resulted in pre-weaning death [42,43]. Importantly, megakaryopoiesis was negatively affected in *BCL2L1*/*Mcl-1*-deficient E12.5 embryos, with fewer and smaller MKs than normal and the presence of haemorrhage. Besides, treatment of adult mice lacking *Mcl-1* in the MK lineage with Bcl-xL inhibitor (ABT-737) triggered MK apoptosis in vivo [42], with a rapid drop in MK and platelet numbers. In summary, these studies revealed that prosurvival Bcl-xL and Mcl-1 are critical for the survival of the MK lineage (Figure 1).

Data on the role of Bcl-xL in human megakaryopoiesis are at present unclear. Contrary to mouse data, some studies have suggested that the intrinsic apoptotic pathway is required for proplatelet formation, with “regional” rather than “whole-cell” death facilitating the necessary cytoskeletal rearrangements. (i) Treatment of human MKs with the pancaspase inhibitor zVad blocked proplatelet formation [72,73]. (ii) Retroviral overexpression of Bcl-2 in human cells leads to impaired proplatelet formation [73]. (iii) A thrombocytopenic family with an activating mutation of *cytochrome c* was shown to have increased differentiation of MKs with an abnormal release of the platelets in the bone marrow [74]. Furthermore, a second mutation of *cytochrome c* was identified in an Italian family with impaired thrombopoiesis [75]. It was reported that the two mutations in *cytochrome c* affected both apoptosis and cellular bioenergetics [75,76] suggesting a potentially complex mechanism underlying cytochrome c mutation and its effects on MK maturation and platelet production. Moreover, a recent murine study has challenged the impact of proplatelet formation in vivo, where MK membrane budding, rather than proplatelet formation was shown to supply the majority of the platelet biomass without induction of cell death [60]. Thus, future studies are needed to gain mechanistic insights into how the described conditions influence platelet production. A recent paper reported a major role for Bcl-xL in human erythropoieisis and hematopoietic stem cells [77]; they also reported that Bcl-xL knockdown negatively impacted human megakaryocyte survival, albeit not significantly. Our unpublished results confirm that Bcl-xL knockdown impair human megakaryocyte survival, at early and later stages of their differentiation (personal data). It can therefore not be excluded that defects in Bcl-xL expression in human MKs could be responsible for unexplained thrombocytopenia or thrombocytosis.

## 6. Conclusions

In summary, the platelet life span in circulation is strictly controlled by a balance between prosurvival Bcl-xL and pro-death Bak and Bax. But the trigger of this pathway during platelet in vivo ageing is yet to be defined. Murine studies show that MK survival is regulated by Mcl-1 and Bcl-xL. Future studies are needed to investigate the role of Bcl-xL in human megakaryopoiesis.

## Figures and Tables

**Figure 1 ijms-21-07591-f001:**
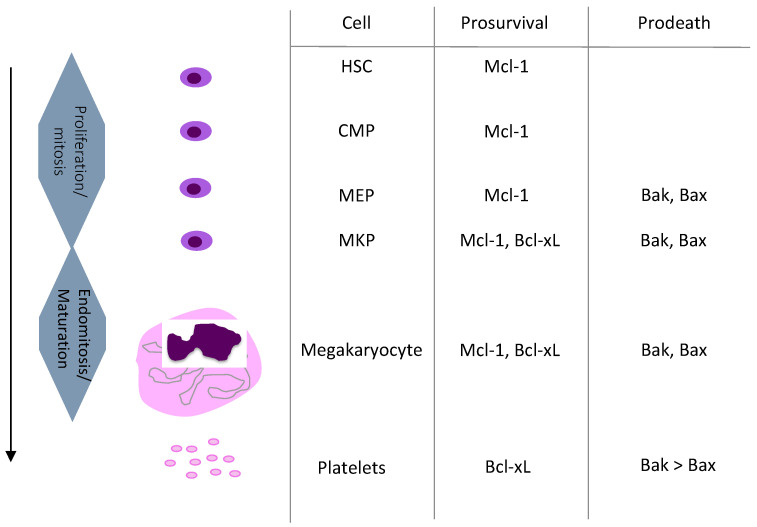
Summary of megakaryopoiesis and current knowledge on the role of the Bcl-xL/Bak axis during megakaryopoiesis. HSC: Hematopoietic Stem Cell. CMP: Common Myeloid Progenitor. MEP: Megakaryocyte Erythroid progenitor. MKP: Megakaryocyte Progenitor.

**Figure 2 ijms-21-07591-f002:**
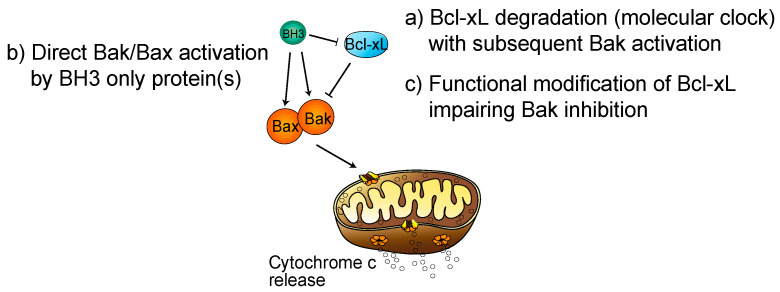
Current hypothesis on the regulation of the balance between Bcl-xL and Bak. Proposed models of Bak/Bax activation in platelets. (**a**) Bcl-xL degrades over time leading to Bak activation (molecular clock model). (**b**) Direct activation of Bak/Bax by BH3 only protein(s). (**c**) Functional modification of Bcl-xL impairing Bak inhibition.

**Table 1 ijms-21-07591-t001:** Causes of thrombocytosis in human.

**Primitive**
Acquired: - Myeloproliferative Neoplasms (Essential thrombocytopenia, Polycythemia Vera, Primitive Myelofibrosis, Chronic Myeloid Leukemia)
- Rarely: Myelodysplastic syndromes (5q-) Constitutional (inherited) thrombocytosis
**Reactive**
Iron Deficiency
Inflammation/Cancer
**Secondary to asplenism**

**Table 2 ijms-21-07591-t002:** Causes of thrombocytopenia in human.

**Due to defective platelet production**
Acquired: - Bone marrow infiltration (by malignant cells in most cases)
- Vitamin deficiency (B12, folates) - Defect in megakaryocyte differentiation (myelodysplastic syndromes) Constitutional (inherited) thrombocytopenia - Absence of megakaryocytes - Defects in megakaryocyte differentiation (mutations in transcription factors) - Defects in proplatelet formation
**Due to increased platelet destruction**
Immunological processes - Immune Thrombocytopenia Purpura - Drug-induced thrombocytopenia
Non-Immunological processes: platelet consumption - Microangiopathies (including thrombotic thrombocytopenia) - Large hemorrhages - Disseminated intravascular coagulation - Sepsis
**Secondary to abnormal pooling** (hypersplenism due to splenomegaly)

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
