# Peer review of "Regulation of Platelet Production and Life Span: Role of Bcl-xL and Potential Implications for Human Platelet Diseases"

_ijms, 2020, doi:10.3390/ijms21207591_

Round 1
Reviewer 1 Report
This is a review of the factors and mechanisms, which control the life span of blood platelets. The review covers both the generation of platelets from megakaryocytes (MK), MK and platelet survival and death both in murine and human systems. The authors present the current evidence that platelet life span is dictated by the balance of the survival factors Bcl-xL and the death factors Bak. However, it is correctly pointed out that the trigger(s) of the survival / death pathways are not well defined yet.
Overall, this is a very nice and well-written review on a topic of wide interest and clinical importance. The authors are clearly experts in this area.
I have only minor suggestions, which the authors may want to consider.
- The process of survival/death is tightly controlled by proteins of the Bcl-2 family, which could have apoptotic and anti-apoptotic properties. However, there are many of these Bcl-2 protein members. Is it certain that only Bcl-xL and Bak are important? Which other family members are expressed in platelets?
- This review does not address well how these Bcl proteins really work at the molecular level. I am aware that this reflects the knowledge in this field. However, it would be of interest for this review if the possible mechanisms of action are briefly mentioned.
- The authors comment that either direct Bak activation (by an unknown factor) or decreased inhibitory effect on Bak could trigger apoptosis. Could this perhaps be due to ubiquitylation? The components of this system are highly expressed in human platelets, and multiple proteins are affected.
Reviewer 2 Report
The article by Josefsson et al. is entitled „The role of Bcl-xL in platelet production and life span: implications for human platelet disease.“ The title matches, however, a little with the content. The structure of the article is confusing. Only p.2 and p.3, row 180 at p.5 and the last paragraph are related to the topic of the article. The rest are general information on platelets and their diseases. In my opinion, the topic of the article is not yet ripe for a review paper. Indeed the number of references (47) includes also references on general issues related to platelets, so the number of reference to the topic is low. Authors alone report „Data on the role of Bcl-xL in human megakaryopoiesis are at present unclear.“ This statement can be indeed extended to the whole article including the role of Bcl-xL and other anti-/proapoptotic molecules in platelet physiology and pathophysiology.
There are also some minor issues:
Figure 1 – authors have not checked the last version of this figure, there are many „?“
Figure 2 is too small and unclear, the quality must be improved
r.36 – „Platelets are present at 150 to 400 million per millilitre (mL) of blood.“ while r.154 – „The normal platelet count ranges from 150 to 450 millions per mL in humans.“ – this again confirms the inhomogeneity of the paper
r.92 – „metronomic degradation“ – unclear expression
p.4 – references should be in numbers, but authors et al. are found at this page
r.155 – „450 G/L“ ?
Reviewer 3 Report
The authors summarized the recent findings on the role of Bcl-xl in regulating platelet life span and megakaryocyte survival. Overall the manuscript is well written and comprehensive. One minor point is thrombocytopenia types may include thrombotic thrombocytopenia and GPIb mutant and deficient-mediated thrombocytopenia.
Round 2
Reviewer 2 Report
The article has been markedly improved and now is almost acceptable for the publication. Here are some minor comments:
r.33-34 - “The inner surface of blood vessels is covered by endothelial cells that act as anticoagulant and serve to maintain blood in its fluid state.” – this is the same “anticoagulant” and “to maintain blood in its fluids state”
r.44 – subscript “2” in thromboxane A2
r.69 – “Caspase” – caspase
r.78-79 – “However, in response to apoptotic triggers such as pharmacological Bcl-xL inhibition with ABT-737 or genetic deletion, both Bak and Bax regulate platelet lifespan in vivo and in vitro” - unclear, rewrite please
r.84 – “Navitoclax” – navitoclax
r.95-96 – “Since there are four A1 genes in mice, another study targeted all A1 isoforms as a unit and platelet counts were still not altered” – rewriting is recommended
Figure 1 – I commented it in my previous review report, authors reacted in the answers but there still the same problems with the figure
r.257 – “Polycythemia Vera, Essential Thrombocythemia and Primary Myelofibrosis” – why upper case letters? There is no reason.
r.261-2 – “Essential Thrombocythemia and Primary Myelofibrosis” – the same as previous comment
r.314 – again the same
r.377 – “Instiute of Medical Research related to Venetoclax” – Institute?, venetoclax
